# Rescue Surgery after Immunotherapy/Tyrosine Kinase Inhibitors for Initially Unresectable Lung Cancer

**DOI:** 10.3390/cancers14112661

**Published:** 2022-05-27

**Authors:** Domenico Galetta, Filippo De Marinis, Lorenzo Spaggiari

**Affiliations:** 1Division of Thoracic Surgery, European Institute of Oncology IRCCS, Via Ripamonti 435, 20141 Milan, Italy; lorenzo.spaggiari@ieo.it; 2Department of Oncology and Hematology-Oncology-DIPO, University of Milan, 20122 Milan, Italy; 3Division of Thoracic Oncology, European Institute of Oncology IRCCS, Via Ripamonti 435, 20141 Milan, Italy; filippo.demarinis@ieo.it

**Keywords:** nsclc, immunotherapy, TKI, surgery, EGFR, ALK, robot

## Abstract

**Simple Summary:**

Locally advanced or metastatic non-small cell lung cancer (NSCLC) has been considered for a long time as an unresectable disease. Chemotherapy was considered the only therapeutic option for these conditions and the results were unsatisfactory. Recent advances in biology and immunology have led to the use of personalized treatments by using tyrosine kinase inhibitors (TKIs) and immune checkpoint inhibitors (ICIs), which produce significant and durable treatment responses. Large trials explored the utility of TKIs and ICIs in neoadjuvant or adjuvant settings, showing good results in terms of radiological response and long-term outcomes. Retrospective case series in patients with the previously unresectable disease who received treatment with TKIs, or ICIs showed important clinical changes that consider the possibility of pulmonary resection of the residual disease. They showed an overall feasibility for pulmonary resection but also raised concerns about the technical challenges. In the present study, we analyzed and reported the surgical and long-term outcomes of patients with initial unresectable, locally advanced, or oligometastatic NSCLC who were treated with TKIs or ICIs achieving a clinical downstaging so as to re-enter resectability.

**Abstract:**

Background: We report the outcomes for unresectable patients with locally advanced or oligometastatic non-small cell lung cancer (NSCLC) treated with tyrosine kinase inhibitor (TKI) or immunotherapy who achieved a clinical downstaging so as to re-enter resectability. Methods: We retrospectively reviewed the clinical, surgical, and pathological data of 42 patients with histologically proven, inoperable NSCLC who received rescue surgery after a good response to TKI or immunotherapy between March 2014 and December 2021. Results: Of 42 patients, 39 underwent pulmonary resection with therapeutic intent (three explorative thoracotomies). There were 26 males, with a median age of 64 years (range, 41–78 years). Twenty-three patients received TKIs and 19 immunotherapies. Anatomic resection was performed in 97.4% of resected patients (38/39) including 30 lobectomies, one right upper sleeve lobectomy, five pneumonectomies, one tracheal sleeve pneumonectomy, and one bilobectomy; a patient underwent wedge resection. Of 10 procedures attempted via a robotic approach, two required conversion to thoracotomy. No intraoperative morbidity/mortality occurred. The median operative time was 190 (range, 80–426) minutes; estimated blood loss was 200 mL (range, 35–780 mL). Morbidity occurred in 13/39 (33.3%). The median length of hospital stay was 6.5 days (range, 4–23 days). Pathologic downstaging was 74.4% (29/39). With a median follow-up of 28.7 months, the 5-year disease-free interval was 46.5%, and the 5-year overall survival was 66.0%; 32/39 patients (82.1%) are alive, 10 with the disease. Conclusions: Lung resection for suspected residual disease after immunotherapy or TKIs is feasible, with encouraging pathological downstaging. Surgical operation may be technically challenging due to the presence of fibrosis, but significant morbidity appears to be rare. Outcomes are encouraging, with reasonable survival during the short-interval follow-up.

## 1. Introduction

Lung cancer is the leading cause of cancer death worldwide. Surgery remains the cornerstone for the treatment of early-stage non-small cell lung cancer (NSCLC). For unresectable or metastatic disease, the treatment of choice for several years has been the use of chemotherapeutic agents that led to dismal results in terms of overall survival [1].

In the last decade, advances in the knowledge of the biology (deep understanding of the epidermal growth factor receptor, EGFR, gene mutations) and immunology (involvement of the immune system) of the tumor cells led to the development of EGFR tyrosine kinase inhibitors (TKIs) and of immune checkpoint inhibitors (ICIs). With the use of TKIs and ICIs, the treatment of advanced and metastatic NSCLC entered an era of co-directed therapy by histology, genotyping, and immunotyping. These new changes in the therapeutic strategies against NSCLC allowed patients to obtain personalized therapies and produce significant and often durable treatment responses [2,3,4,5]. As a result, and with the development of effective TKIs and ICIs drugs, the use of these new agents gained popularity, expanding their use to larger patients’ subsets. On the basis of these studies, numerous trials started to explore the utility of TKIs and ICIs as neoadjuvant or adjuvant therapy in surgically resectable NSCLC, showing benefits in terms of the radiological response of the tumor and survival [6,7,8]. Initial retrospective case series in patients with the previously unresectable disease who received treatment with TKI, or ICI showed important clinical changes that consider the possibility of pulmonary resection of the residual disease. These studies have also shown the overall feasibility of pulmonary resection, but have also raised concerns about the technical challenges and postoperative surgical outcomes for this particular patient population [9,10,11,12,13,14].

The aim of the present study was to analyze and report the surgical and long-term outcomes of patients with initially unresectable, locally advanced, or oligometastatic NSCLC who were treated with TKIs or ICIs achieving a clinical downstaging so as to re-enter resectability.

## 2. Materials and Methods

The study was performed in accordance with the Declaration of Helsinki; the Ethics Committee of our Institution waived the need for ethics approval and the need to obtain consent for the collection, analysis and publication of the retrospectively obtained and anonymized data for this non-interventional study.

This was a retrospective review of our prospectively maintained thoracic surgical database. We identified all the patients who underwent surgical resection for lung cancer after a response to the TKIs or ICIs. All included patients gave written informed consent for participation in this study and for publication of the study data.

All patients evaluated in this study received previously a cytological or histological diagnosis of NSCLC and standard staging procedures and they were judged to be unresectable at the time of presentation, on the basis of (a) an extended mediastinal lymph node involvement or (b) the presence of locally advanced disease or (c) a distant metastatic disease (oligometastatic). Thus, they received TKIs or ICIs therapy as a definitive treatment as the first or second line treatment; they were referred for surgical intervention if the computed tomography (CT) scan or positron emission tomography (PET) scan findings were suggestive of a response to the disease in terms of a reduction of mediastinal lymphadenopathy and/or the primary tumor (Figure 1), or good control of distant metastatic disease (i.e., brain metastasis treated by radiation therapy).

TKIs include the first generation of EGFR-TKI (gefitinib and erlotinib), the second generation (afatinib), the third generation (osimertinib) and the anaplastic lymphoma kinase (ALK) TKI (crizotinib and alectinib). ICIs include the anti-programmed death-1 (PD-1) agents (nivolumab and pembrolizumab), and anti-PD-L1 agent (atezolizumab).

All patients scheduled to undergo pulmonary resection received the following preoperative clinical evaluation: patient history and physical examination, complete cardiac and pulmonary function tests, arterial blood gas analysis, and a quantitative perfusion scan. The preoperative re-staging included a CT scan of the chest, brain, and abdomen, as well as a PET scan. Magnetic resonance imaging of the brain was performed in case of symptomatic or suspicious brain metastasis. Endobronchial ultrasound was preoperatively performed in all patients with a CT or PET scan with suspicious mediastinal lymph node involvement: patients with single station, homolateral N2 disease, persistent after TKI or ICI, were operated on; patients with contralateral nodal involvement (pN3) or those with multiple nodal station involvement were excluded from “salvage resection”. Patients were staged according to the eighth edition of the TNM staging system.

Clinical, surgical and pathological records were individually reviewed. Age, sex, forced expiratory volume in 1 s (FEV1), forced vital capacity (FVC), carbon monoxide lung diffusion capacity (DLCO), histology, type of resection, clinical and pathological stages and pathological responses, operative time (duration from the incision to wound closure), intraoperative blood loss, mortality, complications, intensive care unit stay, hospital stay, and survival were abstracted.

The Kaplan–Meier method was used to estimate the overall survival (OS) and disease-free survival (DFS). OS was measured from the date of surgery until death or loss to follow-up, while DFS was calculated from the first day of the operation until any event, such as tumor recurrence, the incidence of a second cancer, or a secondary condition (i.e., death). Patients were otherwise censored on the date of the last follow up. Survival and tumor recurrence was assessed by patient follow-up with CT of the chest, abdomen, and brain every 4 months for the first 2 years, every 6 months for the following 3 years, and annually thereafter.

## 3. Results

A total of 42 patients who were operated on between March 2014 and December 2021 after receiving a TKI or ICI treatment were identified. The clinical and demographic information of these patients is listed in Table 1. There were 26 males and 16 females with a median age at operation of 64 years (range, 41–78 years). All patients were smokers. The main spirometric values of the cohort are reported in Table 1.

Adenocarcinoma was the main histologic subtype (78.5%, *n* = 33). Clinical stages before TKIs/ICIs therapy included 11 stage IIIa (26.1%), 10 stage IIIb (23.8%), 5 stage IIIc (12.0%), and 16 stage IV (38.1%) (Table 1). Patients with stage IV had their distant metastases cured by radiotherapy or were completely cured by TKI or ICI treatment. The drug regimen included TKIs in 23 patients (54.9%), and ICIs in 19 (45.1%) (Table 1). Fourteen patients (33.3%) received systemic cisplatin-based chemotherapy before TKIs (*n* = 4) and ICIs (*n* = 10), while one patient received navelbine and another one pemetrexed before an ICI drug. No patient received preoperative radiation therapy on the pulmonary o mediastinal lesions. No patient prior to surgery demonstrated resistance to TKIs or ICIS. The median duration from the start of therapy to surgery was 15 months (range, 6–23 months).

Surgical pulmonary resection was accomplished in 39 patients (92.8%); there were three (7.1%) explorative thoracotomies due to the intraoperative evidence of millimetric pleural metastases. Surgical details of the resected patients are reported in Table 2.

Surgery was performed in 28 cases (71.8%) on the right side, and in 11 cases (28.2%) on the left side. An anatomic resection (lobectomy or greater) was performed in 38/39 cases (97.4%). It included lobectomy in 30 (76.9%), right-upper sleeve lobectomy in 1 (2.6%), bilobectomy in 1 (2.6%), pneumonectomy in 5 (12.7%), and tracheal-sleeve pneumonectomy in 1 (2.6%). One patient (2.6%) received a wedge resection. The surgical approach included a lateral muscle sparing thoracotomy in 29 cases (74.4%), and a minimally invasive approach in 10 cases (25.6%) (eight robotic and two VATS). A minimally invasive robotic approach was attempted in 10 cases (all lobectomies) but in two (5.1%) it was necessary to convert to thoracotomy. Intraoperative morbidity and mortality were nil. The median operative time was 190 min (range, 80 to 426 min). The Median estimated blood loss was 200 mL (range, 35 to 780 mL). Three patients (7.7%) required blood transfusion postoperatively.

Complete resection (R0) was achieved in 38 patients (97.4%); only one patient had an R1 (<1 mm) positive margin (2.6%) on the rib margin in the case of a right upper lobe lobectomy with chest wall resection. The overall pathological downstaging was 74.4% (29/39). Nodal downstaging (preoperative pN+ to pN0) was observed in 41.0% (*n* = 16). Three patients moved from pN3 to pN0, 12 patients passed from cN2 to pN0, and one patient passed from cN1 to pN0. A complete pathological response was achieved in nine patients (23.1%).

The postoperative results of the resected patients’ group are summarized in Table 3.

Thirty and 90-day mortality was nil. Complications occurred in 13 patients (33.4%). Two patients (5.2%) experienced a major complication (broncho-pleural fistula and hemothorax, respectively) requiring re-intervention. Eleven patients (28.2%) presented minor complications: five with prolonged air leaks (12.7%); two (5.2%) with vocal cord paresis; three with arrhythmias (7.7%) and one with respiratory failure (2.6%). Only six (15.4%) patients spent 12 postoperative hours in the intensive care unit. The median hospital stay was 6.5 days (range, 4 to 23 days).

After completion of surgical therapy, some patients received some form of adjuvant treatment: two patients received radiation therapy; one patient received systemic chemotherapy, and all the remaining patients resumed previous treatments (TKIs or ICIs).

With a median follow-up of 28.7 months (range, 2–92) of the resected population, 32/39 patients (82.1%) are alive, 10 with disease (seven with extra-thoracic metastasis, and three with local recurrence). Three- and 5-year DFI was 46.5% (median, 16.4 months) (Figure 2),

While 3-and 5-year OS was 79.2% and 66.0%, respectively (Figure 3) (median 26.9 months).

## 4. Discussion

For a long time, sequential or concurrent chemo-radiotherapy has been considered the standard treatment for advanced NSCLC with a dismal prognosis. Standard chemotherapy with platinum-based combinations have been largely used, demonstrating an increase in the overall survival of 8–10 months [1,8]. EGFR-TKI has been proven to be able to convert advanced NSCLC to an operable status after neoadjuvant treatment among patients harboring EGFR mutations, including T790M, L858R, and exon-19 deletion. EGFR-TKI has shown to be extremely effective in more than 70% of advanced NSCLC with EGFR mutations [15]. Several case reports on patients with advanced stage undergoing EGFR-TKI therapy in a neoadjuvant setting suggest that the tumor could be downstaged by the TKI treatment [11,16,17]. Ning [11] demonstrated that some patients with advanced NSCLC and EGFR mutation gained the opportunity to be operated on after treatment with gefitinib: progression free survival was 14 months while overall survival was up to 36 months. Similar to this study, other case reports reported improved overall survival after EGFR-TKI treatment and downstaging of advanced NSCLC to an operable status [18].

In the last decade, immunotherapy has gained popularity, rapidly becoming a pivotal treatment option for patients with advanced NSCLC. The mechanism of action of the ICIs is to block immune inhibitor signals on the cancer surface allowing for the recognition of tumor antigens by the patient’s T lymphocytes and ultimately resulting in the destruction of tumor cells by the immune system. After the good results of the initial study of ICIs in patients with metastatic NSCLC [5], several prospective randomized clinical trials have shown improved survival and tolerability with PD-1 and PD-L1 blockade in selected patients with advanced NSCLC [19,20]. These good results for ICIs in terms of improvement of disease-free survival and overall survival led to different clinical trials for these agents in neoadjuvant settings. The surgical outcomes of these trials are important because they may help to understand the feasibility and safety of lung resection after ICIs. Chaft and colleagues reported the first series of five patients operated on after treatment with ICIs [13] suggesting that the procedure was feasible, but cautioned that mediastinal and hilar fibrosis might develop as a result of a response to the treatment. Bott and colleagues, recently reported, in a series of 19 operated cases after ICIs treatment, that the resections were feasible, although challenging, without undue morbidity [12]. The same data and general impressions were reported by Yang and colleagues [21]. A very recent study by Bott and co-workers [14], represents the largest experience of pulmonary resection after ICIs. In this study the authors reported data on the resection of 20 patients with stage I to IIIA NSCLC after neoadjuvant treatment with two doses of nivolumab: also, in their experience, mortality was nil both intra- and postoperatively while the postoperative morbidity was similar to that reported in previous multicentric trials.

In our experience, despite the technical difficulties encountered during the operation, we had no operative morbidity and mortality. Furthermore, postoperative mortality at 30- and 90 days was nil and this compares favorably with the low mortality rate (0 to 7%) reported in previous trials of neoadjuvant chemotherapy followed by pulmonary resection [22,23,24,25,26]. In addition, the rate of perioperative morbidity in our study was similar to those reported in previous multicentric trials [22,23,24,25,26]. In particular, if compared with the SWOG S9900 trial, which compared chemotherapy followed by surgery with surgery alone in patients with stage IB to IIIA NSCLC, patients in our study showed favorable rates of pneumonia (0% vs. 7%), reintubation (0% vs. 7%), and arrhythmia (5.8% vs. 16%). The rate of prolonged air leak was higher in this study than in the SWOGS9900 (14.2% vs. 9%) [26], and it is unclear whether this is because of the treatment strategies (four were in the TKIs group, and one in the ICIs group) or instead it is related to the small size of the patient cohort. In our series, we had two major postoperative complications: a bronchopleural fistula after a right pneumonectomy, occurring in a patient who received pembrolizumab and was treated with an open window, and a hemothorax in a patient who received gefitinib that was reoperated and without evidence of a clear hemorrhagic foci.

In the present study, 74.3% (29/39) of pulmonary resections were performed through an open approach, very similar to the study by Bott and colleagues (14/20) [14] but compared to this study in which a thoracotomy was started in seven cases (35%), we performed a direct thoracotomy in 27 cases (69.2%). This difference might be explained by the fact that the patients in our study had an initial more advanced disease associated with sizeable tumors with hilar and/or mediastinal nodal disease compared to Bott’s study. We started with a robotic approach in 10 cases, but we had to convert it to a thoracotomy in two cases because of the presence of hilar and perivascular fibrosis. Although some studies have suggested the feasibility of lobectomy by a minimally invasive approach after preoperative chemotherapy in patients with IIIAN2 disease [27,28,29,30], a recent analysis by Krantz, a co-worker at the National Cancer Database, showed that lobectomy was accomplished with a less invasive approach in only 25.7% of such patients [31].

As reported in previous studies, the main type of surgical pulmonary resection was lobectomy, accounting in our experience for 76.9% (*n* = 30) and it varies in other studies from 36% to 77% [12,14,21]. Compared to other previous experiences [11,12,14,21], we had a higher rate of pneumonectomy (12.7%) and we are the first group to report a tracheal sleeve pneumonectomy after immunotherapy. About this last intervention, which was performed with extracorporeal membrane oxygenation assistance, we would like to report that it was a very difficult procedure due to an extended fibrotic scar involving the trachea-bronchial angle and the superior vena cava, requiring an associated resection and reconstruction of the vena cava by a handmade bovine pericardium. The postoperative course of this patient was uneventful and he was discharged on the eighth postoperative stay and was well and without disease in a 33-month follow-up.

An important point in the surgical resection of these patients is the amount of fibrosis after these therapies. The pathological features of the lesion and the lymph nodes suggested the replacement of tumors by fibrotic scar tissue, and the concentration of focal residual tumors was limited in areas of fibrous stroma and lymphocyte infiltration (Figure 4).

On the other hand, it is a common experience that neoadjuvant chemotherapy may downstage the advanced NSCLC, but the scar tissue and impaired physical condition may lead to a higher rate of surgical complications and mortality. From a pathological point of view, TKIs and ICIs will act on the blood supply and lead to the necrosis of the tumor and the formation of scar tissue; this tissue may be a challenge for the surgery, increasing the risks of intraoperative morbidity. According to our experience, the resection of lesions has been difficult for robotic procedures leading to conversion in two cases. However, in some cases, both after TKIs and ICIs, we encountered serious difficulties in the isolation of the vessels, in the dieresis of the fissure and also in the lymph node dissection due to the high rate of fibrosis and scarring at the level of the vascular tissues (Figure 5).

We have no data to confirm whether the length of preoperative therapy may have affected long-term outcomes.

We would like to emphasize, in accordance with Bott and colleagues [14], that also in our study, the radiological assessment of treatment response with CT might not be accurate after immunotherapy when used in a neoadjuvant setting. In fact, the overall pathological response was identified in 16/19 (84.2%) patients receiving immunotherapy whereas post-treatment CT scans most commonly showed a partial response in 10, and stable disease in nine. On the contrary, for patients receiving TKIs, the overall pathological response was 56.5 (13/23), and there was a greater correspondence between post-treatment CT scans, and pathological responses: CT scans showed a major radiological response in 12 cases, a partial response in 6, and a stable disease in 5. The nine patients in the current study with a complete pathological response (23.1%) (six receiving immunotherapy, three TKI), radiologically showed a major radiological response in three, a partial response in four and stable disease in two. This good rate of complete response after immunotherapy was recently confirmed by the CheckMate 816 (NCT02998528) study, a randomized, phase 3, open-label study evaluating NIVO+chemo versus chemo as neoadjuvant therapy for resectable NSCLC.32. In fact, this randomized trial showed a statistically significant improvement in the pathological response with neoadjuvant NIVO+chemo vs. chemo alone (24.0% vs. 2.2%, *p* < 0.0001) [32] confirming the curative potential of immunotherapy.

Although the long-term outcome was not one of the endpoints of this study, the survival figure of our series of operated patients after immunotherapy or TKIs is very encouraging. In fact, with a median follow-up of 28.7 months (range, 2–92 months), the 5-year projection of survival is 66.0%; 32/39 patients (82.1%) are alive, 10 with disease; of these, three patients had local recurrence cured by radiation therapy, and seven patients had distant metastasis: two with bone metastasis and five with brain metastases treated by radiation therapy. Seven patients died: three patients due to causes not related to the tumor (one for COVID-19 infection, one for acute respiratory distress syndrome, and one for acute ischemic stroke; the other four patients died due to disease progression. Of these four patients, two had squamous cell carcinoma, one had adenocarcinoma, and one had a sarcomatoid tumor. From a surgical point of view, these four patients underwent, in three cases, an open lobectomy and a left pneumonectomy. Recurrence occurred in the brain in two cases, the contralateral supraclavicular node in one case, and the ipsilateral supraclavicular node + chest wall in the last case, and all were treated with radiation therapy. We speculate that the low deaths by metastasis during the followed period could be due to the effectiveness of TKI and ICI treatments that were administered, even after surgery, thus ensuring disease control.

Patients who relapsed (10 alive with disease and four died with disease) demonstrated resistance to TKIs/ICIs (*n* = 14, 35.9%). Specifically, in 10 patients who received TKIs (three erlotinib, five gefitinib, two afatinib), and four who received ICIs (three pembrolizumab, one Nivolumab).

Comparing the 5-year survival rate of patients operated on for locally advanced NSCLC (29.8%) [33], or with oligometastatic disease (29.4%) [34,35] or who underwent salvage surgery (5-yr survival, ranging from 20% to 78%) [36], the survival data of our series appear really encouraging (66.0%).

This study has several limitations: (a) although clinical and surgical data were extracted from our prospectively maintained database, the study remains retrospective in nature, and therefore, subject to bias in regard to data collection; (b) the experience of pulmonary rescue surgery after TKIs/ICIs treatments is still limited. Although our study is the largest one on this topic in the literature, the number of patients in this analysis is relatively small.

Finally, in this small study, we demonstrated the feasibility of performing surgical resection after TKIs and ICIS treatments, reporting good results in terms of short and long-term postoperative outcomes. The use of TKIs and ICIs in the treatment of locally advanced or metastatic NSCLC is constantly being increased in clinical practice and it is very likely that surgeons will be asked more and more to perform pulmonary resections on an increased number of patients who have received these effective drugs. Thus, from a surgical point of view, it is interesting to report the surgical outcomes under these new therapeutic strategies.

## 5. Conclusions

In conclusion, in light of our experience, we confirm that lung resection for suspected residual disease after ICIs or TKIs is feasible, with encouraging pathological downstaging. Surgical operation may be technically challenging due to hilar and vascular fibrosis, but significant morbidity appears to be rare. Short- and long-term outcomes are encouraging, with a reasonable survival rate.

## Figures and Tables

**Figure 1 cancers-14-02661-f001:**
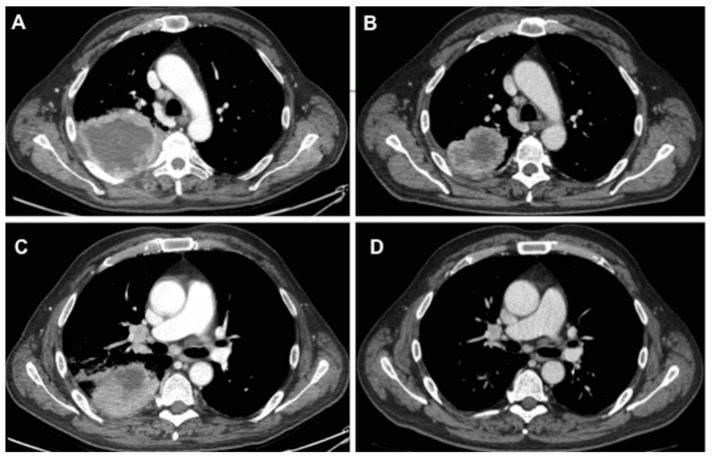
Computed tomography scan showing the initial radiological aspect of tumor and lymph nodes (**A**,**C**) compared to the aspect after treatment with pembrolizumab (**B**,**D**).

**Figure 2 cancers-14-02661-f002:**
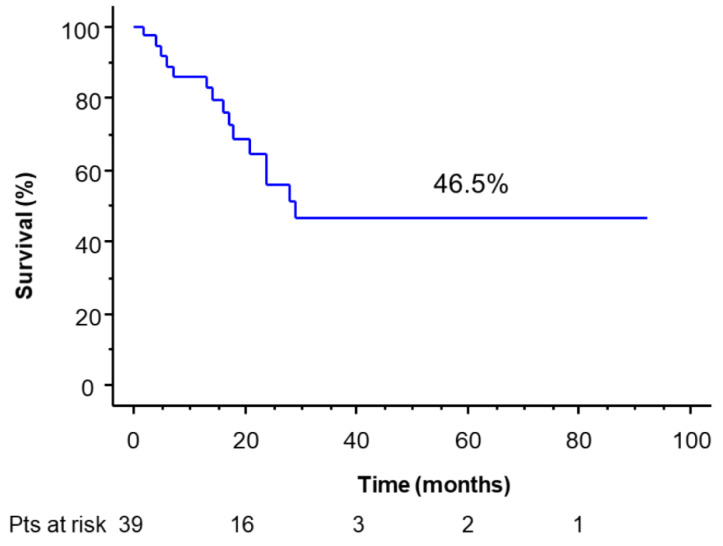
Disease free interval of the operated patients.

**Figure 3 cancers-14-02661-f003:**
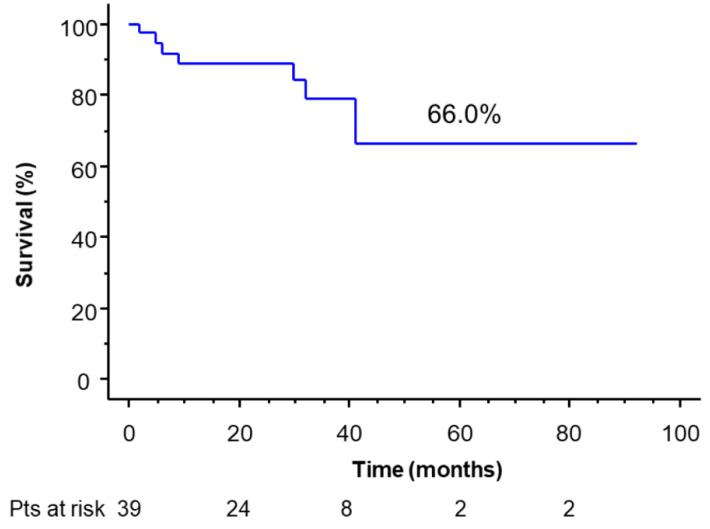
Overall survival of the operated patients.

**Figure 4 cancers-14-02661-f004:**
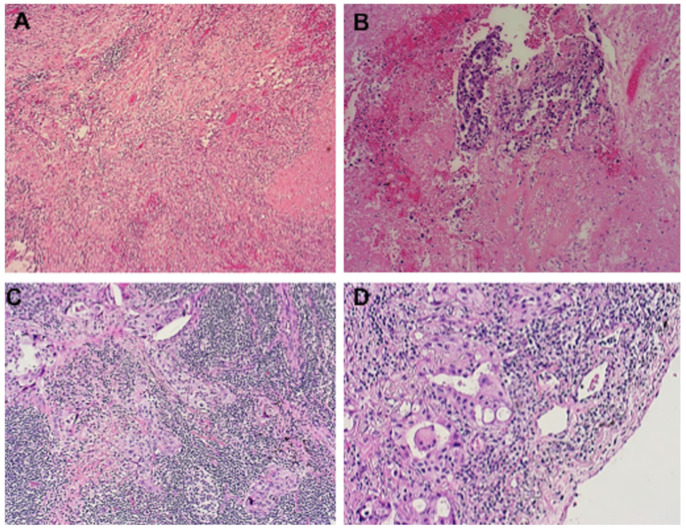
Microphotograph showing the histological features in patients who received ICIs (pembrolizumab, (**A**) *hematoxylin & eosin, H&E x100,* and (**B**) *H&E x250*) or TKIs (erlotinib, (**C**) *H&E x100* and (**D**), *H&E x400*). In both cases (**A**,**C**), it is evident the presence of massive fibrotic scar tissue with high concentration of lymphocyte infiltration, and the presence of rare tumor foci (**B**,**D**).

**Figure 5 cancers-14-02661-f005:**
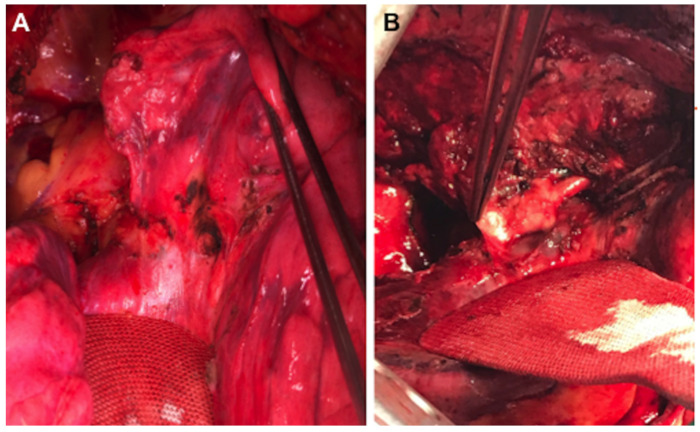
Intraoperative view of two operated cases after treatment with erlotinib (**A**) and pembrolizumab (**B**). A. Presence of diffuse fibrotic tissue in the fissure. B, Presence of perivascular fibrosis and scars making more difficult vessels isolation.

**Table 1 cancers-14-02661-t001:** Baseline characteristics of the study group (*n* = 42).

Characteristics	No. (%)
Median Age (range), y	64 (41–78)
Sex	26M/16F
Smoking history	
Former or current	42 (100)
FEV1 (mean % predicted ± SD)	78.64 ± 14.32
FVC (mean % predicted ± SD)	81.58 ± 15.58
DLCO (mean % ± SD)	79.42 ± 14.86
Histology (%)	
Adenocarcinoma	33 (78.5)
Squamous cell carcinoma	5 (11.9)
Adenosquamous	2 (4.8)
Sarcomatoid	2 (4.8)
Clinical stage before TKIs/ICIs	
IIIA	11 (26.1)
T1cN2	2 (4.8)
T2aN2	2 (4.8)
T2bN2	3 (7.0)
T4N0	4 (9.5)
IIIB	10 (23.8)
T3N2	8 (19.0)
T4N2	2 (4.8)
IIIC	5 (12.0)
T1bN3	1 (2.4)
T1cN3	1 (2.4)
T3N3	2 (4.8)
T4N3	1 (2.4)
IV	16 (38.1)
IVa	14 (33.3)
IVb	2 (4.8)
Drug regimen	
TKIs	23 (54.9)
Gefitinib	5 (12.0)
Erlotinib	4 (9.5)
Afatinib	6 (14.2)
Osimertinib	5 (12.0)
Crizotinib	2 (4.8)
Alectinib	1 (2.4)
ICIs	19 (45.1)
Pembrolizumab	14 (33.3)
Nivolumab	3 (7.0)
Atezolizumab	2 (4.8)
Months from start therapy and surgery (median, range)	15 (6–23)

FEV1 = forced expiratory volume in 1 s; SD = Standard Deviation; FVC = forced vital capacity; DLCO = carbon monoxide lung diffusion capacity; TKIs = tyrosine kinase inhibitors; ICIs = immune checkpoint inhibitors.

**Table 2 cancers-14-02661-t002:** Surgical results of the resected patients (*n* = 39).

Characteristics	No. (%)
Type of resection	
Lobectomy	30 (76.9)
Bilobectomy	1 (2.6)
Right-upper sleeve lobectomy	1 (2.6)
Pneumonectomy	5 (12.7)
Tracheal sleeve pneumonectomy	1 (2.6)
Wedge resection	1 (2.6)
Side	
Right	28 (71.8)
Left	11 (28.2)
Surgical approach	
Open	29 (74.4)
Robotic	8 (20.5)
VATS	2 (5.1)
Conversion from robotic to open	2 (5.1)
Intraoperative morbidity and mortality	0
Median operative time (range), min	190 (80–426)
Median estimated blood loss, mL	200 (35–780)
Complete resection (R0)	38 (97.4)
Overall pathological downstaging	29 (74.4)
Nodal pathological downstaging (pN2-1 to pN0)	16 (41.0)
Complete pathological response	9 (23.1)

VATS = Video-assisted thoracic surgery.

**Table 3 cancers-14-02661-t003:** Postoperative results of the resected patients (*n* = 39).

Variable	No. (%)
30-day mortality	0
90-day mortality (%)	0
Overall morbidity (%)	13 (33.4)
Major	2 (5.2)
Broncho-pleural fistula	1 (2.6)
Hemothorax	1 (2.6)
Minor	11 (28.2)
Prolonged air leak	5 (12.7)
Vocal cord paresis	2 (5.2)
Arrhythmia	3 (7.7)
Respiratory failure	1 (2.6)
ICU stay, median (range), days	0 (0–1)
Hospital stay, median (range), days	6.5 (4–23)

ICU = intensive care unit.

## Data Availability

Available upon request.

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
