# Peer review of "Rescue Surgery after Immunotherapy/Tyrosine Kinase Inhibitors for Initially Unresectable Lung Cancer"

_cancers, 2022, doi:10.3390/cancers14112661_

Round 1
Reviewer 1 Report
The work presented by Galetta et al is interesting and might have high clinical relevance in NSCLC patient management.
They showed that after TKI treatment (either in EGFR mutated or EML-ALK translocated patients) or immune check point inhibitors treatment, showed a clinical downstaging allowing surgery. This is particularly interesting because the primary tumors on the analyzed cohort were unresectable before the therapeutic treatment. The data is in accordance with several previous studies acknowledged by the authors but I think their study merit to be published in Cancers because of the greater number of patients compared to previous studies.
I have only minor points that I hope will help the authors to make even better their study.
1) I would like to see some more information regarding the low deaths by metastasis during the followed period in the survival analysis. Do the authors think that this could be due to the ICI/TKI treatment? Could it be a bias in the chosen patient cohort? Some information or at the very least, some comments in the discussion would be highly appreciated.
2) Plots in Figures 2 and 3 are with commas instead of points for the decimals, please revise.
3) Up to line 230 in the discussion it is only about previous published studies. On the contrary in this point the authors start to talk about their work but without clearly mentioning. Then, in line 246 they start for the first time one paragraph in the discussion with “In the present study” as it could be the start point to discuss their own data. Please revise all these parts since it is very confusing.
Author Response
Reviewer #1
We thank Reviewer #1 for considering our work as interesting and for the possible high clinical relevance in NSCLC patient management.
Reviewer #1 reported only minor points:
Question #1:
I would like to see some more information regarding the low deaths by metastasis during the followed period in the survival analysis. Do the authors think that this could be due to the ICI/TKI treatment? Could it be a bias in the chosen patient cohort? Some information or at the very least, some comments in the discussion would be highly appreciated.
Answer #1
We re-evaluated all follow-up data of the patients and we modified (in blue) along the manuscript (at the end of Result section, and in the Comment section, before the limitation paragraph, page 12) the number of patients who are alive with disease.
In the Discussion Section, we have also included (in blue), as required, additional information about the patients who died for disease during the followed period in the survival analysis answering to the question of Reviewer #1.
Question #2
Plots in Figures 2 and 3 are with commas instead of points for the decimals, please revise.
Answer #2
We thank Reviewer #1 for this suggestion and we modified the two Figures as suggested.
Question #3
Up to line 230 in the discussion it is only about previous published studies. On the contrary in this point the authors start to talk about their work but without clearly mentioning. Then, in line 246 they start for the first time one paragraph in the discussion with “In the present study” as it could be the start point to discuss their own data. Please revise all these parts since it is very confusing.
Answer #3
We thank Reviewer #1 for suggesting that we begin the Discussion Section by reporting directly the results of our study.
We partially agree with him because we believe that, as reported at the beginning of the paragraph, an introduction on the state of the art on the new personalized treatments for lung cancer that led us to carry out this study is of fundamental importance.
We modified the first part of the Discussion Section in order to reduce as much as possible overlapping sentences with previous papers.
We start discussing our results starting (in blue) from the line 231 and considering the first lines of the Discussion section as an important introduction for the discussion itself.
Reviewer 2 Report
This study is a retrospective review of the surgical and long-term outcomes of patients with initial unresectable, locally advanced or oligometastatic NSCLC who were treated with various TKIs or ICIs achieving a clinical downstaging so as to re-enter resectability. The result showed that lung resection for suspected residual disease after ICIs or TKIs is feasible, with encouraging pathological downstaging, even though operations can be technically challenging due to hilar and vascular fibrosis but significant morbidity appears to be rare during short-interval follow-up.
Minor consideration
1. Patients received TKIs or ICIs therapy as definitive treatment as first or second line treatment but they were referred for surgical intervention if had computed tomography (CT) scan or positron emission tomography (PET) scan findings suggestive of response of the disease in terms of reduction of mediastinal lymphadenopathy and/or the primary tumor. It will be interesting to analyze how long TKIs or ICIs administrated before surgery and administration duration affected the outcome of the patients? Also, I wonder there is no patient who gets resistance to the TKIs or ICIs before or after surgery, and with relapse?
2. There was no proper control to compare with these data. Even if there was no proper control, it will be interesting to see the outcome of the patient in this paper compared with locally advanced or oligometastatic NSCLC patients who were not treated with various TKIs or ICIs, but referred surgery, if dada existed since these patients were considered as an unresectable.
3. 4-5 pages line 135 – 147 in the results is redundant, which is well summarized in table 1. It will be these sentences to cut short.
4. There was some punctuation marks errors ex) last sentence of simple summary
Author Response
Reviewer #2
We thank Reviewer #2 for his/her comments and suggestions.
Reviewer #2 reported the following minor consideration:
Question #1
Patients received TKIs or ICIs therapy as definitive treatment as first or second line treatment but they were referred for surgical intervention if had computed tomography (CT) scan or positron emission tomography (PET) scan findings suggestive of response of the disease in terms of reduction of mediastinal lymphadenopathy and/or the primary tumor. It will be interesting to analyze how long TKIs or ICIs administrated before surgery and administration duration affected the outcome of the patients? Also, I wonder there is no patient who gets resistance to the TKIs or ICIs before or after surgery, and with relapse?
Answer #1
We thank Reviewer #2 for his timely comments.
As reported in Table 1 and along the manuscript (before Table 2), the median time from the start of the therapy to surgery was 15 months (range, 6-23 months).
About the question if the administration duration may affect the outcome of the patients, we have no data to confirm whether the length of preoperative therapy may have affected long-term outcomes. This concept has been added (in blue) in the Discussion paragraph. We can instead confirm, as reported in the section of the Discussion (lines 282-302), that the prolonged use of these new drugs determines an increase in fibrosis in the place of the neoplastic tissue and at the level of the vascular sheaths.
We would like to thank Reviewer # 2 for pointing out an important aspect: resistance to TKI / ICIs. We reassessed our entire patient series and can confirm that no patient prior to surgery demonstrated resistance to TKIs or ICIS. This concept has been inserted (in blue) in the Results section, at line 150-151.
Conversely, patients who relapsed (10 alive with disease and 4 died with disease) demonstrated resistance to these drugs (n = 14, 35.9%). Specifically, 10 patients who received TKIs (3 erlotinib, 5 gefitinib, 2 afatinib), and 4 who received ICIS (3 pembrolizumab, 1 Nivolumab). This concept has been inserted (in blue) in the Discussion section, at line 343-345.
Question #2
There was no proper control to compare with these data. Even if there was no proper control, it will be interesting to see the outcome of the patient in this paper compared with locally advanced or oligometastatic NSCLC patients who were not treated with various TKIs or ICIs, but referred surgery, if dada existed since these patients were considered as an unresectable.
Answer #2
We thank Reviewer #2 for pointing out this important aspect of the study. This study lacks a control group because its main objective was to evaluate and report the results of a very particular patient population (unresectable patients with locally advanced or oligometastatic NSCLC treated with TKIs or immunotherapy who achieved a clinical downstaging so as to re -enter resectability). However, comparing the 5-year survival rate of patients operated on for locally advanced NSCLC (29.8%) [33], or with oligometastatic disease (29.4%) [34] or who underwent salvage surgery (5-yr survival, range from 20% to 78%) [35], survival data of our series appear re-ally encouraging (66.0%). These updates have been reported (in blue) in the Discussion section, lines 346-349.
Question #3
4-5 pages line 135 – 147 in the results is redundant, which is well summarized in table 1. It will be these sentences to cut short.
Answer #3
We thank Reviewer #2 for this suggestion. We have shortened these sentences (in blue).
Question #4
There was some punctuation marks errors ex) last sentence of simple summary.
Answer#4
We thank Reviewer #2 for this observation. We provided to correct them.